

# 1 Impact of two high resolution altimetry mission concepts for
# 2 ocean forecasting

Mounir Benkiran[1], Pierre-Yves Le Traon[1,2] , Elisabeth Rémy[1] and Yann Drillet[1]
[1]Mercator-Ocean International, 31400 Toulouse, France
[2]Ifremer, 29280 Plouzané, France
*Correspondence*: Mounir Benkiran (mbenkiran@mercator-ocean.fr)
**Keywords:** Data Assimilation; Ocean Forecasting; Surface Water Ocean Topography; Satellite Altimetry;
Observing System Simulation Experiment

## 12 Abstract

Observing System Simulation Experiments (OSSEs) with the Mercator Ocean/Copernicus Marine global 1/12°
data assimilation system has been carried out to compare and quantify the expected performance of two high
resolution altimetry mission concepts envisioned for the long-term evolution (post-2032) of the Copernicus
Sentinel-3 topography mission. The two mission concepts are a constellation of two wide-swath altimeters and a
constellation of 12 nadir altimeters. These two configurations greatly improve ocean forecasting and monitoring
capabilities. Compared to a constellation of three nadir altimeters (the present configuration), analysis and forecast
errors are reduced by a factor of 2. Our results also show that a constellation of two wide-swath altimeters has
better performance than a constellation of 12 nadirs. Compared to a constellation of 12 nadirs, the error of the Sea
Surface Height (SSH) forecast of a two wide swath constellation will be reduced by 14% overall. Improvements
are also observed when analyzing surface currents and Lagrangian diagnostics. A constellation of two wide-swath
altimeters thus seems to be a very promising concept for the long-term evolution of the Sentinel-3 topography
mission.

## 27 1. Introduction

The Copernicus Marine Service implemented by Mercator Ocean International (MOi) provides operational,
regular, and systematic reference information on blue/white/green ocean states for the global ocean and European
regional seas (Le Traon et al., 2019; Le Traon et al., 2021). More than 55,000 expert downstream services and
users are connected to the service. The Copernicus Marine Service responds to public and private user needs and
supports policies related to all marine and maritime sectors.
The Copernicus Marine Service depends on the upstream satellite and in-situ observation infrastructure (Le Traon
et al., 2019). Sea Surface Height (SSH) measurements from satellite altimetry plays a prominent role (see Le Traon
et al., 2017) and the quality of Copernicus Marine analyses and forecasts are directly related to the status of the
altimeter constellation. Copernicus Marine model present (e.g. 1/12° at the global scale and 1/36° at the regional
scale) and future resolutions (e.g. 1/36° at the global scale and 1/108° at the regional scale) currently impose strong
constraints on the altimeter constellation. The observation capabilities lag behind model resolutions. This calls for
a much-improved space/time sampling of the ocean by satellite altimetry.





Wide-swath altimetry (WiSA) that will be demonstrated with the Surface Water Ocean Topography (SWOT)
mission (Morrow et al., 2019; Benkiran et al., 2021; Tchonang et al., 2021) can address these needs. Benkiran et
al. (2022) demonstrated, in particular, that a constellation of two wide-swath altimeters could dramatically improve
the quality of analyses and forecasts. An alternative approach is to fly an optimized constellation of a large number
(e.g. at least 10) of nadir altimeters.
In this paper, we address the relative merits of these two approaches: a constellation of two wide-swath altimeters
and a constellation of 12 nadir altimeters. Results for this global study are presented and discussed in this paper,
which is organized as follows: the WiSA concept is presented in Section 2; Section 3 details the OSSE
methodology; results are discussed in Section 4, and Section 5 provides the main conclusions and
recommendations of the study.

## 14   2. The WiSA concept

The WiSA concept was developed as part of a Phase A study conducted by CNES and industry as an interim
follow-up to the SWOT mission. The objective of the WiSA concept was to leverage the major improvement of
the SWOT swath altimeter with significant modifications to better meet the needs of operational oceanography
and hydrology, while making the satellite simpler, smaller, and more affordable than the SWOT precursor mission.
The WiSA concept is well described in Benkiran et al. (2022).
The so-called WISA #A orbit was selected by CNES using the methodology of Dibarboure et al. (2018) to
maximize the sampling for 1 to 3 swath altimeter satellites (or swath/nadir hybrid constellations). This sun-
synchronous orbit has an altitude of approximately 750 km (14+7/17 revolutions per day) and the altimeter swath
covers latitudes up to 82°, with an exact repeat of 17 days.

## 25   3. OSSE approach

### 26   3.1 Ocean Model

The MOi global ocean forecasting system (see Lellouche et al., 2018), which delivers forecast products for the
Copernicus Marine Service, is used in this study. As described in the literature (Errico et al., 2013), OSSEs use
two different models or model configurations. In our study, we use the same NEMO 1/12° resolution model
(Nucleus for European Modelling of the Ocean, Madec, 2016) but with different configurations and forcings. A
free NEMO3.6 ocean model simulation is used (Nature Run, hereafter referred to as NatRun) to represent the real
ocean and simulate all the synthetic observations for the study. A detailed description and validation of the NatRun
is presented in Benkiran et al. (2021). A second model configuration, which is embedded in the assimilation system
used to assimilate synthetic observations from the NatRun, is based on a NEMO3.1 model configuration, which is
less energetic. This difference in energy is mainly due to the types of forcing used for each of the simulations, as
mentioned in Benkiran et al. (2021). The NatRun uses absolute wind and a less diffusive scheme compared to the
free and assimilated run, which uses relative wind (50%).

### 38   3.2 Simulation of observations and noise

All data (pseudo data) used in our OSSEs are simulated from the same NatRun simulation over a period of 15
months (from October 1, 2014 to December 31, 2015). We simulated the high-resolution sea surface temperature
(1/10°, based on the ODYSSEA L3S SST (https://doi.org/10.48670/moi-00164) coverage), the temperature and
salinity profiles (based on the CORA database (https://doi.org/10.17882/46219) coverage), the ice concentration
data and the Sea Surface Height (SSH). The aim is to stay fairly close to our operational system (see details in
Benkiran at al., 2012). SSH data were simulated along the tracks of 3 nadirs, 12 nadirs and along the swath covered
by 2 wide-swath altimeters (S1 and S2).
The 3 nadir altimeters correspond to the existing baseline operational altimeter constellation (Sentinel-6 (or
Jason3), Sentinel-3A, Sentinel-3B). A constellation of 12 nadir altimeters with SAR capabilities involves 12
altimeters in the same orbiting plane as the current Sentinel-3. The WiSA constellation consists of two wide-swath



altimeters (S1, S2) with their nadirs. The along-track nadir altimeter data were extracted from NatRun at the 1 Hz frequency corresponding to a spatial resolution of 6-7 km from hourly mean SSH fields of the NatRun. A random noise of 2 cm was added to along-track data to take into account altimeter measurement noise (i.e., close to the 1 Hz error budget of the nadir altimeter of the SWOT satellite). For the two-swath altimeters, the WiSA #A orbit (S1) selected by CNES (Dibarboure et al., 2018) was used together with a second (S2) on the same orbital plane, separated by a 180° angle on the orbit circle. All SSH data were simulated from the NatRun using the Jet Propulsion Laboratory's (JPL) SWOT Simulator (Gaultier et al., 2016). The simulator constructs a regular grid based on the baseline orbit parameters of the satellite. The simulator models the most significant errors that are expected to affect the data, i.e., the KaRIn (Ka-band Radar Interferometer) noise, roll errors, phase errors, baseline dilation errors, wet troposphere and timing errors.

In this study, the KaRin error takes into account the calculated wave height, unlike the previous study (Benkiran et al., 2022) where the wave height was fixed at 2m. We use the error of the wet troposphere residual error after correction with 2 beams calculated by the simulator according to the predefined spectrum.

Figure 1 A shows the SSH from the NatRun at a given central date of our 7-day assimilation cycle (analysis window) over the Gulf Stream region. Data coverage along the tracks of the three nadir altimeters over the 7-day analysis window is shown in Figure 1 B while Figure 1 C shows the coverage of the combination 12 nadir altimeters and Sentinel-6, and Figure 1 D of two Wide-Swath Altimeters and Sentinel-6. It can be observed that with two Wide-Swath Altimeters the ocean is almost totally covered by the measurements over a 7-day time period.

**3.3 Data Assimilation scheme**

In this study, an updated version of MOi's data assimilation scheme (SAM2: Système d'Assimilation Mercator V2, Lellouche et al. (2018) and Benkiran et al. (2021)) was used in place of the one used in previous studies (Benkiran et al., 2022). The most important improvement is made on the adaptivity scheme used in the analysis. This gives more weight to the assimilated data and helps to reduce more efficiently the observation residuals. Several improvements and adaptations of this system were made for this study. In particular, a four-dimensional (4D) version of the assimilation scheme is used, in which the analysis uses a 4D subspace and produces a daily model correction of the SSH, temperature, salinity and velocity fields. These corrections are based on 7-day innovations (observations minus model forecasts), weighted by the distance in time from the analysis date. The analysis increment is injected into the model using the Incremental Analysis Update (IAU) method (Bloom et al., 1996; Benkiran and Greiner, 2008). All these updates and their impact on system performance are described in Benkiran et al. (2021). These updates to the system allow us to take better account of the impact of mesoscale structures and uncontrolled temporal frequencies in the 3Nadirs simulation.

**3.4 Experimental set-up**

Starting from the simulated data obtained from the NatRun run, three global OSSEs were carried out using a different NEMO configuration but the same spatial resolution of 1/12° (~ 7 km). An additional experiment was performed, called the Free Run (FR), in which no observation is assimilated. This simulation is used to assess the performance of the assimilative experiments. The data assimilated in the different OSSEs are detailed in Table 1.

|  | SST, IC and T&S Profiles | S3A | S3B | S6 | 12 Nadir S3 | 2 Wide-Swaths |
|---|---|---|---|---|---|---|
| Free Run |  |  |  |  |  |  |
| 3Nadirs | YES | YES | YES | YES |  |  |
| 12Nadirs | YES |  |  | YES | YES |  |
| 2Swaths | YES |  |  | YES |  | YES |

**Table 1.** OSSEs experimental set-up. The rows show the names of the relevant experiments, whereas the columns detail the observations considered in the analysis. The first Column describes the name of the experiment (OSSE). Column 2 shows the standard observations (Sea Surface Temperature (SST), Ice concentration (IC) and Temperature and Salinity profiles (T&S Profiles)). The nadir altimeters considered in the OSSEs: Sentinel-3A

 

(S3A), Sentinel-3B (S3B), Sentinel-6 (S6, Jason3) and 12 nadir altimeters in the same orbital plane as Sentinel-3
(S3). The last column shows the two wide-swath altimetry data (2 Wide-Swaths).
Our simulations start from a free model state on October 1, 2014. A three-month simulation (until the end of
December 2014) was carried out with assimilation of SSH along the nadirs (3Nadirs) together with SST, IC and
T&S data with the same design as the Nadir OSSE presented in Table 1. The final state of this simulation is used
to initialize the different experiments presented in this paper. This allows us to avoid the spin-up period in our
experiments. All the experiments start from the same state on January 1, 2015 and cover one year.
**4. Results**
Analysis of the impact of assimilation of the SSH from the nadir altimeters (12Nadirs) and from the two wide-
swath altimeters is detailed in this section. This analysis is based on comparison between each experiment and our
real ocean (NatRun) data over a period of one year (2015).
We present the results as follows: first, the impact of the different altimeter constellations on the quality of the
SSH analysis and prediction. We then show the impact on the space-time scales by analyzing the spectra
(coherence, error spectrum) of the SSH on boxes (black boxes in Figure 2) of high variability. Scale separation
has been done respecting the spectra of each field to better define the cut-off of the scales (<200 and <500km here)
on which the impact of the assimilation of swath data is greater compared to the Nadir data. We also show the
impact on surface and depth velocities (model dynamics) and system mass (temperature and salinity profiles).
**4.1 Impact on SSH analyses and forecasts**
The aim of this study is to show the impact of assimilating SSH data from 2 wide-swath altimeters compared to a
12-nadir constellation in the same plane as Sentinel-3. The SSH variance in the NatRun computed over one year
(2015) is shown in Figure 2. It shows a high variability in the more energetic regions such as the Gulf Stream (GS)
and Kuroshio (KS) regions. The SSH variance in the NatRun compares very favorably with the estimation from
real altimeter observations (as detailed in Benkiran et al., 2021).
The temporal evolution of SSH variance error over the global ocean for each experiment is shown in Figure 3.
This variance of the error (VarError) is calculated by comparing each OSSE analysis or forecast fields with the
NatRun as shown in Benkiran et al. (2022). This variance decreases over a few weeks (6 weeks) to reach a stable
state for the analyses (continuous lines) and the forecasts (dotted lines). The experiment assimilating 2 Wide-
Swath and S6 SSH hereafter referred to as the 2Swaths experiment, has an SSH error variance of about 9.6 cm$^2$ in
analysis compared to the experiment assimilating 12 Nadirs, hereafter referred to as the 12Nadirs experiment,
which has an error variance of 11.0 cm$^2$. The gain is of about 14% for the analysis and 15% for the forecast (dotted
lines). We summarize these overall statistics in Table 2. We also show the impact on different ranges of scales. As
we are trying to highlight the impact on different scales, we have calculated statistics on SSH errors with scale
separation based on the spectra of these errors. We show a gain of about 4% for scales smaller than 200 km.
In **Figure 4**, we compare the variance of each experiment with that of the 3Nadirs simulation. In the first figure,
we have the global map of SSH analysis error variance of the 3Nadirs run. In the middle figure we have the
difference between the SSH error variance of the 3Nadirs experiment and that of the 12Nadirs. The red areas
correspond to the improvement areas. We have a clear improvement with 12Nadirs in the western edge currents
and the Antarctic Circumpolar Current (ACC). Overall, we have 70% of the improvement points on the global
ocean. In the same way, we have an additional improvement with the assimilation of wide swath data (2Swaths,
bottom figure) compared to the 12Nadirs run (78% of improvement points on the global ocean).
**Figure 5** summarizes the main results of these analyses. The three panels represent the mean error variance as a
function of latitude for the total error, the error for wavelengths smaller than 500 km and the error for wavelengths
smaller than 200 km. Assimilation of data from the two wide-swath altimeters reduces the error at each latitude
(red curves on the panels). This improvement is more pronounced at middle and high latitudes than at low latitudes.
The impact on western boundary currents and Antarctic Circumpolar Current (ACC) is more evident at mesoscale
(<200 km) as a function of the amplitude of the signal. This answers our question: data assimilation of SSH from
two wide-swath altimeters provides a better estimate of the mesoscale than nadir altimeters.



The aim of our study is also to show the impact of wide swath data assimilation on the forecast skill for different lead times. A comparison of the forecast error is made on the three experiments (3Nadirs, 12Nadirs and 2Swaths) at different timescales between 1 and 7 days. A large impact is observed in the ocean forecast (Figure 6), where the SSH forecast variance error over the global ocean increased by about twice between the 1st and the last (seventh) day. The assimilation of 12Nadirs (blue line), is reduced by almost 80% compared to the 3Nadirs experiment (with 3 nadirs, black line). With the 2Swaths experiment (red line) we have an additional gain of 10% for the 1-day forecast and 14% for the 7-day forecast. The 7-day forecast error for 2Swaths is of the same order as the 2-day forecast error for the 3Nadirs experiment. This gain in the 7-day forecast shows us that the model is able to keep following more of the mesoscale structures introduced by the analysis than the 12Nadirs experiment.

**4.2 Spectral analysis and coherence**

Statistics and wavenumber Power Spectral Density (PSD) and spatial and temporal coherence for each OSSE compared to the Nature Run are discussed in this section for 2 regions, the Gulf Stream and Kuroshio areas.

At the top of Figure 7 we show the SSH variance error of the 3Nadirs and the differences of the 12Nadirs and 2Swaths experiment compared to the 3Nadirs simulation and the temporal evolution of SSH RMS errors in the Gulf Stream area (red box in Figure **2**). In Figure 7 B (2Swaths-3Nadirs), we have an important difference (negative) compared to the difference between 12Nadirs and 3Nadirs (Figure 7 C). With 2Swaths, the error is smaller than with Nadir. This is clear from the graph showing the evolution of SSH RMS error in Figure 7 D (red line). The evolution of the RMS of this error in SSH of the 2Swaths experiment (red curve) is lower than that of 12Nadirs except at the beginning of the year, and presents an annual average of 0.080 cm instead of 0.086 over this region of the Gulf Stream area.

Figure 8 shows the power spectra of the SSH error in a variance preserving form (Thomson and Emery, 2014) in the Gulf Stream area. The assimilation of nadir altimeter data from 12 nadirs (blue curves) reduces the error at different scales compared to the 3Nadirs experiment (black line) for wavelengths in excess of 180km. On the other hand, with the 2Swaths experiment (red curve), this error is much reduced compared to the 12Nadirs (blue curve) from 150km onwards. A temporal spectral analysis (Figure 8 B) is also carried out to highlight the impact of assimilating swath data at different scales in relation to NatRun. Coherence is defined as the correlation between two signals as a function of wavelength (Benkiran et al., 2022). The effective temporal resolution for the 12Nadirs and 2Swaths is around 30 days (blue and red curves) compared with the 3Nadirs (50 days, black curve) with a coherence of 50%. There is better resolution with 2Swaths (red curve) compared to 12Nadirs at all wavelengths.

Looking at a different region, at the top of Figure 9 we show the SSH variance error in the Kuroshio area of the 3Nadirs and the differences of the 12Nadirs and 2Swaths experiment compared to the 3Nadirs simulation. On the right-hand figure, we have the difference between 12Nadirs and 3Nadirs. This confirms that even in high energy regions, the analyzed SSH in the 2Swaths experiment is better than in the 12Nadirs experiment. We confirm this by the figure below, which represents the evolution of this SSH RMS error as a function of time over the year 2015. The red line, corresponding to the 2Swaths experiment, has on average over the whole period a lower mean compared to the blue line, corresponding to the 12Nadirs experiment. Figure 10 shows on the left the Power spectra of the SSH error with respect to the NatRun, while on the right we have time spectral coherence with respect to the NatRun. These spectra are calculated on the black box defined on Figure 9. Over this region, the impact is negligible, with an improvement for scales below 200km and a slight degradation compared to the 12Nadirs for above 200km on this box. Similarly, a very small difference regarding the temporal coherence is found between the 12Nadirs and 2Swaths experiments.

The less energetic Kuroshio region (with low errors) is fairly well controlled by the assimilation of Nadirs data (12Nadirs), whereas the Gulf Stream region (with higher errors) needs the assimilation of data from 2Swaths to constrain these errors. The Figure 11 shows a comparison of the wavenumber-frequency energy spectra over these two regions of the SSH NatRun (Benkiran et al., 2022). In the Kuroshio region (figure on the right), the NatRun has a relatively low energy at all spatial scales (especially between 10 and 150 days) compared with the Gulf Stream region (figure on the left). This confirms the impact of 2Swaths data assimilation over these two regions.





### 4.3 Impact on temperature, salinity and zonal velocities

Figure 12 shows the variance and mean of the temperature and salinity error as a function of depth for the global ocean. This error (innovation for assimilation) is calculated between the temperature and salinity profiles simulated from the NatRun and their model equivalents in each OSSE. The temperature error profile shows a maximum value at about 100m depth, which corresponds to the thermocline. This error is significantly reduced by the assimilation of 12 Nadirs (blue profile) compared to 3Nadirs (black profile). The assimilation of wide-swath altimetry data does not degrade this score and we even have a slight improvement between 100 and 750m depth. For salinity (right figure), the improvement is less clear, but no degradation is observed regardless of the depth.

In Figure 13, we compare the evolution of the variance of the velocity analysis error (NatRun - OSSEs) as a function of time over the year 2015 at the surface for the zonal (U, top) and meridional (V, bottom) components. We observe a reduction that sets in after one month and remains constant over the year. As shown in Table 3, we have a reduction of these errors for the global ocean (in variance) of about 10% for the 2Swaths experiment compared to the 12Nadirs experiment. Figure 14 similarly shows the average variance and mean error both for the zonal (U) and meridional (V) velocity as a function of depth for the global ocean for each of the experiments. The SSH assimilation from 12Nadirs (blue profiles) shows a good reduction of the error on the two velocity components on the global ocean mean and on the whole water column compared to the 3Nadirs simulation. The 2Swaths (2Wide-Swaths) experiment (red profiles) brings a significant reduction of this error compared to 12Nadirs (blue profiles) on the water column.

### 4.4 Lagrangian diagnostics

The aim of this study is to quantify the contribution of assimilating swath data compared with 12Nadirs data on the quality of the model's currents (analysis and forecast). With this new generation of swath altimeters, we should have a better representation of mesoscale phenomena. To quantify this impact, we carried out the same analysis as that presented in the paper by Tchonang et al. (2021). We evaluate the ability of OSSEs to reproduce the particle drift observed with the NatRun compared to our OSSEs. Lagrangian particle positions are initialized uniformly at 1/12° (between latitudes 66°S and 66°N, the maximum latitude of the Jason/S6 orbit). These particles are advected by the surface currents in each simulation (NatRun and OSSEs) over a period of 7 days. We used 6,130,410 particles in each experiment. We used the Ariane software (Blanke and Raynaud, 1997) to carry out these experiments.

To highlight the impact of the different missions, we compared the arrival distance of these particles after 7 days with their positions in the NatRun at the same date. The separation distance for the 3Nadirs run is shown in Figure16 (**Figure16** A). In this figure, we can see that after 7 days, particles in areas of high activity (western boundary currents, tropics, etc.) are more than 100km from their position in the NatRun. The figures show the change in separation distance compared with 3Nadirs. Figures 15 B and C show respectively the change in direction of particles in the 12Nadirs and 2Swaths experiments compared with the 3Nadirs run. The 2Swaths experiment (Figure C) improves particle trajectory compared with the 12Nadirs run (Figure B) over almost the entire globe. On average, 2Swaths shows a gain of 17.5% compared with 3Nadirs, while 12Nadirs shows a gain of 14.46% (Table **5**). This table also shows (first column) the percentage of particles that remain at a distance of less than 50km from their position in NatRun (our reality). In the 2Swaths experiment, 73.9% of particles are 50km from reality (NatRun), compared with 64.5% for 12Nadirs and 40.6% for the 3Nadirs run.

An analysis by different energy zones was carried out to better show the impact of swath data compared to nadirs data. We chose to carry out an analysis in three bands of Eke (Eddy kinetic energy) to distinguish regions with different regimes. Figure 17 shows the result of this analysis. Regions with Eke greater than $0.04 m^2/s^2$ (Figure 17 A) show a gain of 11% with 2Swaths compared to 12Nadirs (63.4% for 2Swaths instead of 52.3% for 12Nadirs). In the second energy range ($0.01 m^2/s^2 < Eke < 0.04 m^2/s^2$, Figure B) the gain remains fairly significant at around 10%. On the other hand, in the low-energy zones ($< 0.01 m^2/s^2$, weaker currents) the gain is less marked, but remains significant at around 6% in these regions of low variability. Table 5 summarizes these percentage gains for the 2Swaths experiment in relation to the 12Nadirs overall for each region with a different energy range. This



leads us to conclude that the impact is positive on surface currents in analysis and forecasting. The assimilation of
swath altimeter data has a positive impact on our analysis and forecasting system compared with nadir data.
**5. Summary and conclusions**
The SWOT mission, which was launched in December 2022, will likely demonstrate the major contribution of
swath altimeters to ocean monitoring and forecasting. OSSEs (Benkiran et al., 2021; Tchonang et al., 2021) carried
out with a global 1/12° data assimilation system has shown that SWOT will provide a large improvement in ocean
analyses and forecasts but with a limitation due to its 21-day time revisit. A complementary study (Benkiran et al.,
2022) showed, however, that the constellation of two wide-swath altimeters should provide a much larger
improvement.
To confirm and quantify the expected performance, new OSSEs (with the same system setup as the previously
mentioned ones but with refined observation error characteristics) were carried out in this study. The OSSEs
compared the relative merits of a constellation of two wide-swath altimeters and a constellation of 12 nadir
altimeters (in the same orbital plane as Sentinel-3). Such configurations are envisioned by ESA for the long-term
evolution (post-2032) of the Copernicus Sentinel-3 topography mission to meet the requirements expressed by the
Copernicus Marine Service and its applications (CMEMS, 2017). These two configurations greatly improve ocean
forecasting and monitoring capabilities. Compared to a constellation of three nadir altimeters (the present
configuration), analysis and forecast errors are reduced by a factor of 2. Our results also show that a constellation
of two wide-swath altimeters performs better than a constellation of 12 nadirs. Compared to a constellation of 12
nadirs, the error of the SSH forecast of a two wide-swath constellation will be reduced by 14% overall.
Improvements are also observed for surface currents (~10%) and Lagrangian diagnostics.
Flying a constellation of two wide-swath altimeters thus looks to be a very promising solution for the long-term
evolution of the Sentinel-3 constellation and the Copernicus Marine Service. End-to-end simulations taking into
account a full error budget and all processing steps (e.g. reduction of large scale errors, intercalibration) before
data are assimilated would be needed to fine tune these results. It will be useful, in particular, to assess how along-
track long wavelength errors (e.g. due to orbit error, tidal or inverse barometer correction errors) by inducing
spurious cross-track errors impact the ability of a constellation of multiple altimeters to map the mesoscale signals.
With respect to swath techniques, the comparison of these results with real data from SWOT will also allow us to
verify the degree of realism of our simulations.

**Competing interests:** The contact author has declared that none of the authors has any competing interests
**Acknowledgments:** The study was funded by ESA and was also carried out as part of a partnership agreement
between Mercator Ocean International and CNES.



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



# 1 Figures

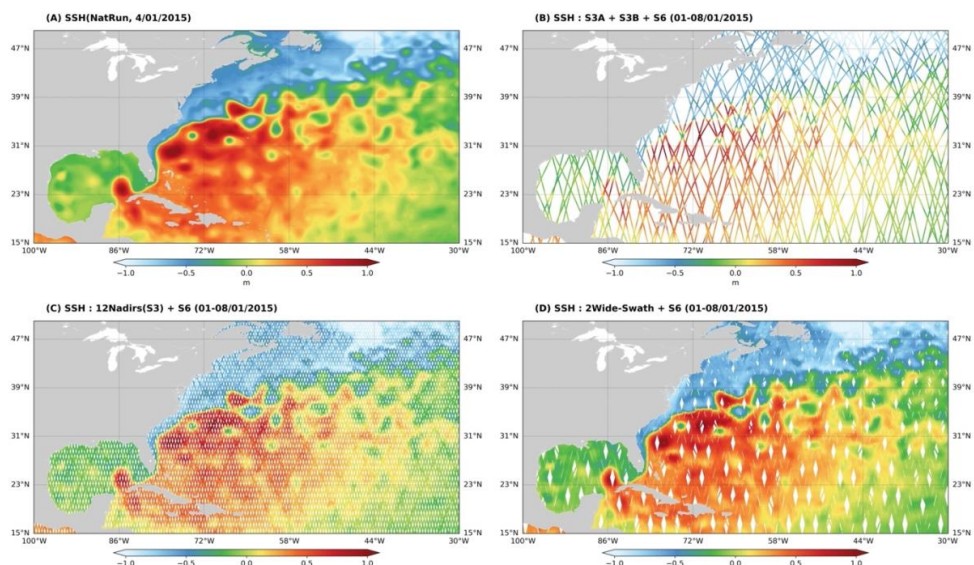

Figure 1: (A) SSH from Truth Run (**NatRun**) on 4th January 2015, (B) simulated along-track data from Sentinel-6, Sentinel-3A and Sentinel-3B (3Nadirs) for seven-day assimilation cycle, (C) simulated along-track data from 12 nadir altimeters and Sentinel-6 (12Nadirs) in the same orbital plane as Sentinel-3 and (D) 2Wide-Swath and Sentinel-6 (2Swaths) data (01/01/2015-08/01/2015).

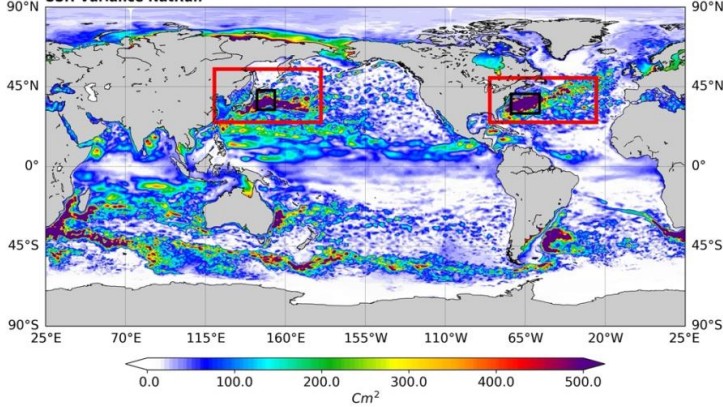

Figure 2: SSH variance (in cm$^2$) in the NatRun over the period from February to December 2015. The red boxes denote the rectangular sub-regions for which statistics were calculated and the black boxes those for which wavenumber spectra and coherence analyses were performed.

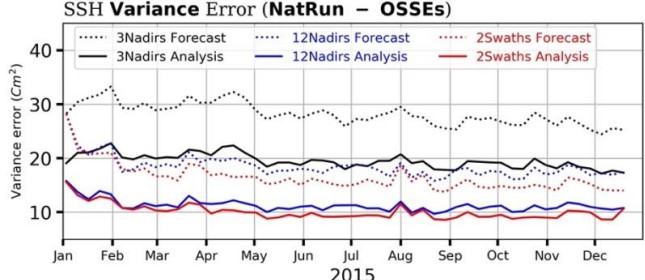

Figure 3: The temporal evolution of the SSH error variance in global ocean analysis and forecast over 2015. Results
obtained by comparing the SSH ocean analysis (solid lines) and forecast (dash lines) with the SSH from the
NatRun. Experiments 3Nadirs: black lines, 12Nadirs: blue lines and with 2Swaths: red lines (see Table 1 for
descriptions of each experiment).



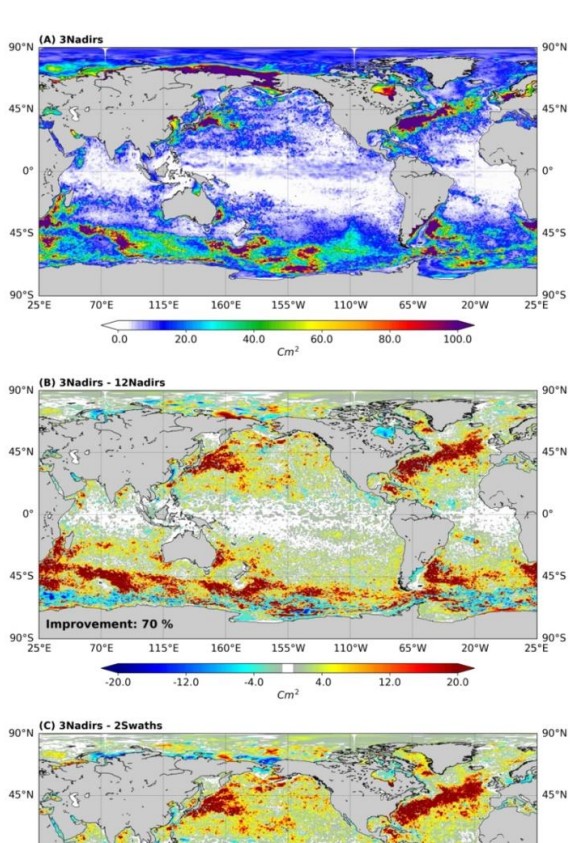

Figure 4: Global Maps of SSH analysis error (NatRun – Model Analysis) variance (in cm$^2$, 2015). (A) 3Nadirs,
(B) difference between analysis error variance of 3Nadirs and 12Nadirs, (C) difference between analysis error
variance of 3Nadirs and 2Swaths.

| | SSH error variance (cm$^2$) | | | |
|---|---|---|---|---|
| | Analysis | Forecast | Analysis (WL < 500Km) | Analysis (WL < 200Km) |
| 12Nadirs | 11.0 | 18.0 | 9.1 | 10.2 |
| 2Swaths | 9.6 | 15.7 | 84 | 9.8 |
| Gain | 14% | 15% | 8% | 4% |

Table 2: SSH ocean analysis and forecast error statistics during the year 2015. Columns 2 and 3 represent the
analysis and forecast variance of error computed from the difference between the OSSE and the NatRun (VarError,
cm$^2$). Columns 4 and 5 show the ratio of the variance of the error relative to the 3Nadirs Run variance (Var*, %).



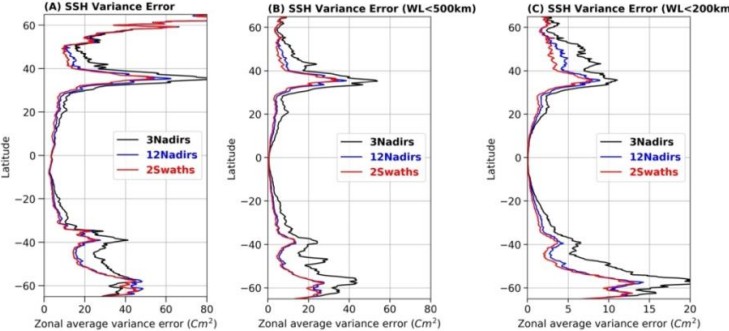

Figure 5: The zonal SSH averaged error variance: (A) for full scales, (B) for scales less than 500 km and (C) for scales less than 200 km; assimilation of 3Nadirs Run (black lines), 12Nadirs Run (blue lines) and 2Swaths Experiment (red lines). Units are cm$^2$.

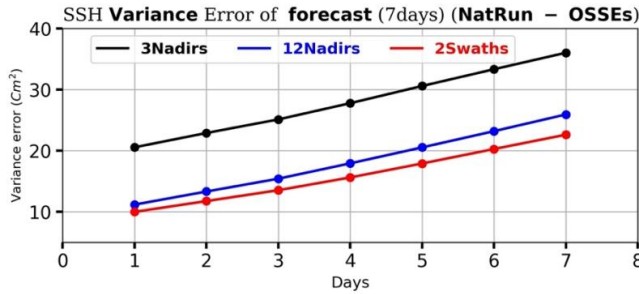

Figure 6: Variance of the error for each day of forecast (7 days, cm$^2$) considering the SSH on the Global Ocean, over the period from February to December 2015. Results obtained by comparing the SSH of the ocean forecast in the 3Nadirs (black line), 12Nadirs (blue line) and 2Swaths (red line) experiments with the data from the NatRun.



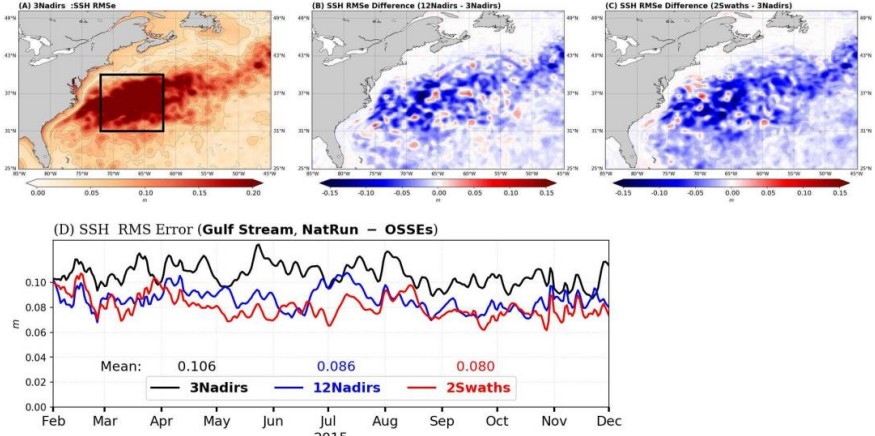

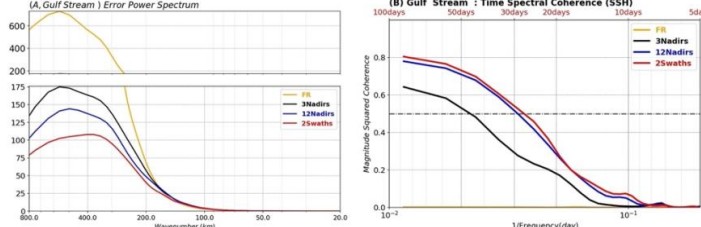

Figure 7: The SSH RMS error over 2015 for the 3Nadirs run (left, top) and the difference in RMS error compared to the 3Nadirs for the 12Nadirs (middle, top) and 2Swaths (right, top) for Gulf Stream area (red box Figure 2). The temporal evolution of the SSH RMS error analysis for this area; experiments 3Nadirs: black line, 12Nadirs: blue line and with 2Swaths: red line.

Figure 8: Power spectra SSH error with respect to the NatRun; the spectra are shown in a variance-preserving form (cm2), (A) Gulf Stream area (black box in Figure 7), (B) Time spectral coherence with respect to the NatRun (same area), 3Nadirs experiment: black line, 12Nadirs experiment: blue line and 2Swaths experiment: red line.



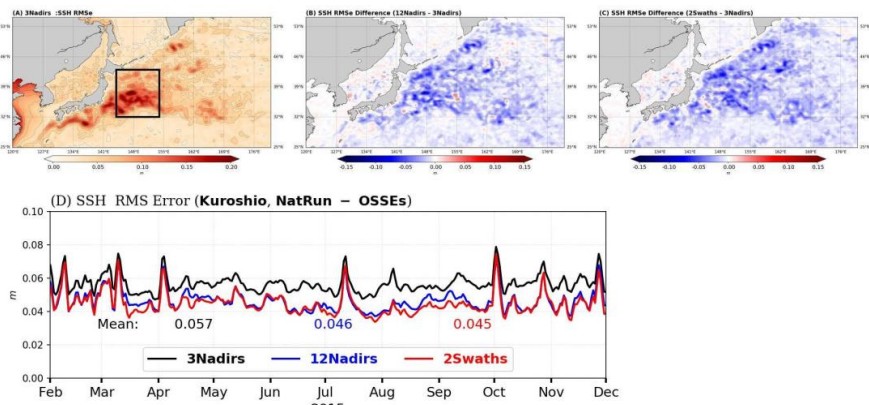

Figure 9: The SSH RMS error over 2015 for the 3Nadirs run (left, top) and the difference in RMS error compared
to the 3Nadirs for the 12Nadirs (middle, top) and 2Swaths (right, top) for Kuroshio area (red box Figure 2). The
temporal evolution of the SSH RMS error analysis for this area; experiments 3Nadirs: black line, 12Nadirs: blue
line and with 2Swaths: red line.

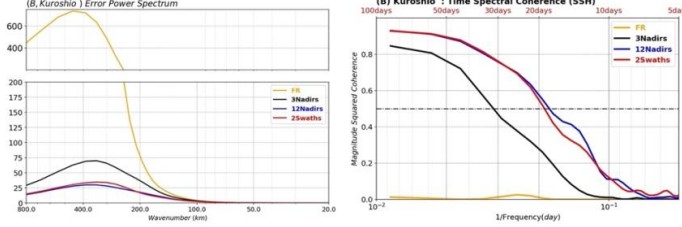

Figure 10: Power spectra SSH error with respect to the NatRun; the spectra are shown in a variance-preserving
form (cm²), (A) Kuroshio area (black box), (B) Time spectral coherence with respect to the NatRun (same area),
3Nadirs experiment: black line, 12Nadirs experiment: blue line and 2Swaths experiment: red line

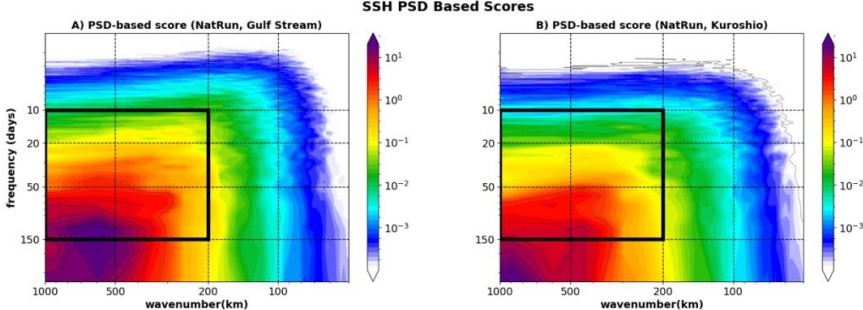

Figure 11: SSH NatRun wavenumber-frequency energy spectra in Gulf Stream box (black box in **Figure** 8, left
panel) and Kuroshio (black box in Figure 9, panel right).





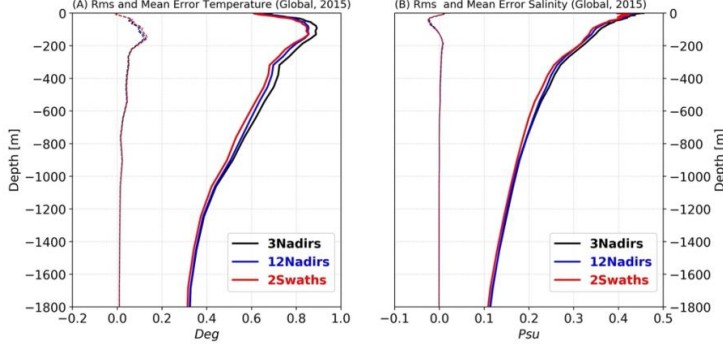

Figure 12: Global averaged RMS error (solid lines) and mean (dashed lines): (A) temperature (in Deg) and (B)
salinity (Psu) over the period 2015. The results were obtained by comparing temperature and salinity profiles of
OSSEs with the profiles simulated from NatRun; 3Nadirs (black lines), 12Nadirs (blue lines) and 2Swaths (red
lines).

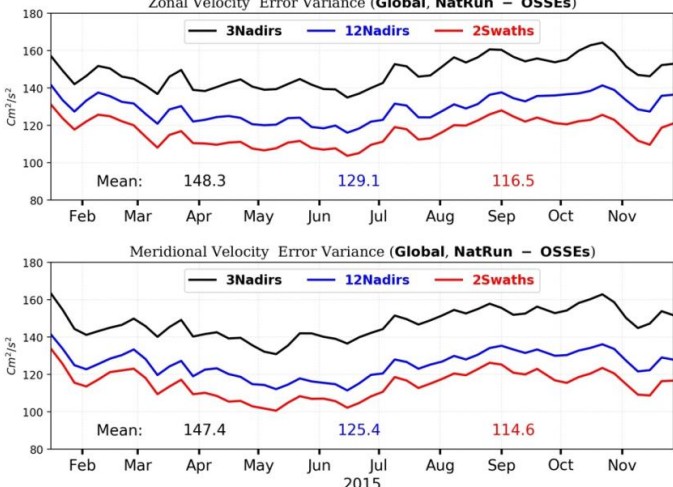

**Figure 13**: Temporal evolution of zonal (upper figure) and meridional velocity (lower figure) error variance
($cm^2/s^2$) for 7-day ocean analysis over the period from January to December 2015. The results were obtained by
comparing both components of the velocity (U; V) to that of the NatRun. Black curves with assimilation of 3
Nadirs (3Nadirs); blue curves for 12Nadirs experiment and red curves with assimilation of 2Swaths experiment (2
Wide-Swath).

|  | Analysis error variance ($cm^2/s^2$) | |
|---|---|---|
|  | Zonal velocity | Meridional velocity |
| 12Nadirs | 129.1 | 125.4 |
| 2Swaths | 116.5 | 114.6 |
| Gain | 11.2% | 9.6% |

Table 3: Velocity Ocean analysis error statistics during the year 2015. Column 1 represents the analysis zonal
velocity and Column 2 the analysis meridional velocity for the 12Nadirs and 2Swaths experiments; variance of
error computed from the difference between the OSSE and the NatRun (VarError, $cm^2/s^2$). Last line (Gain) shows
the ratio of the variance of the error relative to the 12Nadirs Run variance (Var*, %).



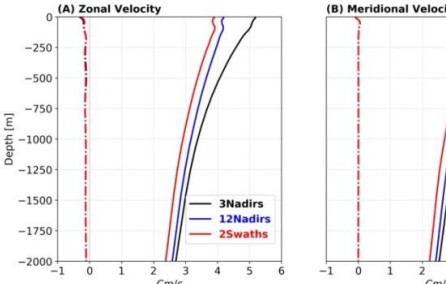

Figure 14: Global averaged RMS error (solid lines) and mean (dashed lines): (A) zonal velocity (in cm/s) and (B) meridional velocity (cm/s) over the period 2015. The results were obtained by comparing temperature and salinity profiles of OSSEs with the NatRun; 3Nadirs (black lines), 12Nadirs (blue lines) and 2Swaths (red lines).

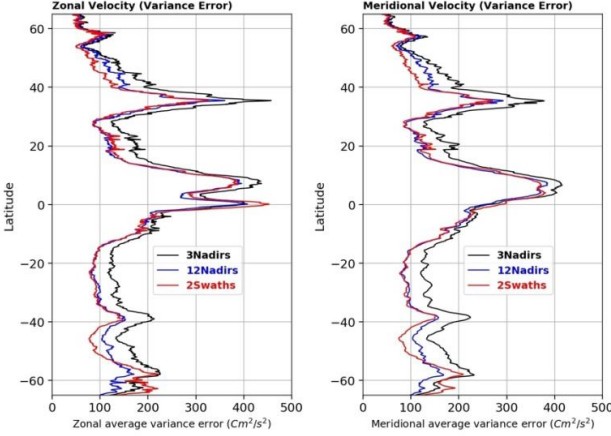

Figure 15: The average error variance of zonal (left figure) and meridional velocity (right figure): assimilation of 3Nadirs (black lines) and assimilation of 2Swaths (red lines). Units are cm$^2$/s$^2$.



|  | Variance of the error for each day of forecast (7 days, cm$^2$) | | | | | | |
|---|---|---|---|---|---|---|---|
|  | Forecast for Day 1 | Forecast for Day 2 | Forecast for Day 3 | Forecast for Day 4 | Forecast for Day 5 | Forecast for Day 6 | Forecast for Day 7 |
| 3Nadirs | 20.5 | 22.8 | 25.1 | 27.7 | 30.5 | 33.3 | 36.0 |
| 12Nadirs | 11.1 | 13.3 | 15.4 | 17.9 | 20.5 | 23.1 | 25.9 |
| 2Swaths | 9.9 | 11.7 | 13.5 | 15.6 | 17.8 | 20.2 | 22.6 |
| Gain: 2Swaths/12Nadirs | 11.9% | 13.3% | 13.9% | 14.7% | 14.7% | 14.4% | 14.6% |

Table 4: Variance of the error for each day of forecast (7 days) considering the SSH on the Global Ocean, over the
period from February to December 2015.

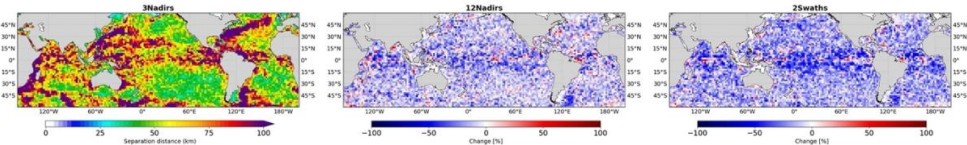

Figure16: (A) Separation distance between particles deployed in 3Nadirs (three nadirs) and their equivalent in the
NatRun after 7 days of advection by their respective surface current velocities. The results were averages in 2x2
degree bins. (B, C) Changes (in %) of the separation distance when using 12Nadirs or 2Swaths (2Wide-Swaths)
surface velocities relative to results with 3Nadirs.

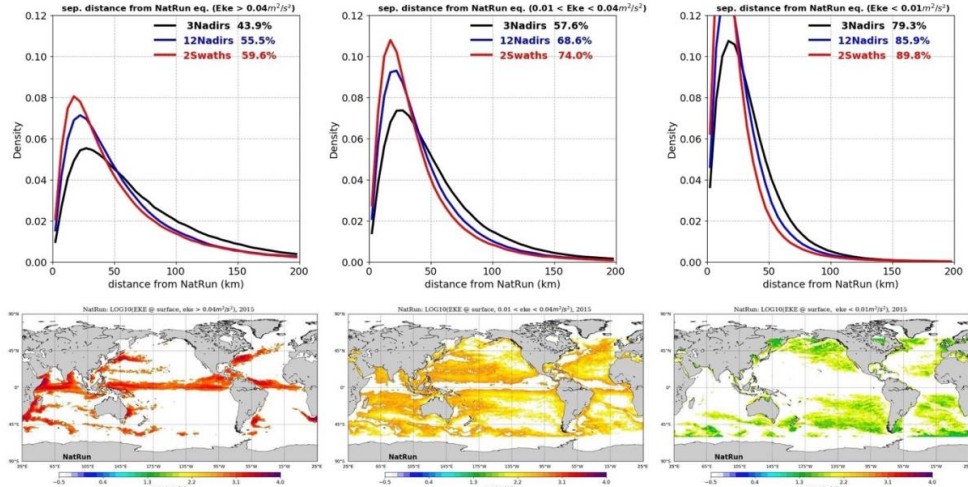

Figure 17: Binomial histogram of separation distance for all OSSEs: Each graph represents a region with a range
of NatRun Eke (mean annual Eke): (A) region with high Eke (>0.04 m$^2$/s$^2$), (B) region with medium Eke (0.01
m$^2$/s$^2$ < Eke > 0.04 m$^2$/s$^2$) and (C) region with low Eke (<0.01 m$^2$/s$^2$). The figure noted for each experiment
represents the percentage of the number of particles for which the distance between the OSSE particles and their
NatRun equivalent remains less than 50Km.

| Runs | Global | | Eke > 0.04m$^2$/s$^2$ | | 0.01m$^2$/s$^2$ < Eke < 0.04m$^2$/s$^2$ | | Eke < 0.01m$^2$/s$^2$ | |
|---|---|---|---|---|---|---|---|---|
|  | % < 50km | Change relative to 3Nadirs | % < 50km | Change relative to 3Nadirs | % < 50km | Change relative to 3Nadirs | % < 50km | Change relative to 3Nadirs |



| | | | | | | | | |
|---|---|---|---|---|---|---|---|---|
| 3Nadirs | 40.6 | - | 28.1 | - | 35.2 | - | 52.8 | |
| 12Nadirs | 64.5 | 14.46% | 52.3 | 17.10% | 65.7 | 14.08% | 73.2 | 10.94% |
| 2Swaths | 73.9 | 17.50% | 63.4 | 19.25% | 75.8 | 17.30% | 79.5 | 14.33% |

1  Table 5: Summary of statistics for the separation distance of OSSE Lagrangian particles with their NatRun
2  equivalent after 7 days of advection, for the global ocean and three regions with a range of NatRun Eke (see Figure
3  17).