# Peer review of "Impact of two high resolution altimetry mission concepts for"

_EGUsphere, 2024_

## Referee Comment (RC2)

[General comments]

This paper investigates the impacts of two swath and 12 nadir altimeters on the analysis and forecast accuracies conducting the OSSEs. The topic appears to be interesting, however, there are critical issues described below.

Firstly, the contents are largely similar to those of a previous paper by Benkiran et al. (2022), but with only the addition of the results assimilating 12 nadir altimeters. To establish the novelty of this paper, I strongly suggest conducting a wider variety of experiments to comprehensively investigate how many nadir altimeters are beyond 1- and 2-swath altimeters and cannot substantially increase the accuracy.

Secondly, in the OSSE, data assimilation experiments should not employ true values from the Nature Run. However, this study uses true values for the initial conditions of assimilation experiments, which is inconsistent with the OSSE protocols. Therefore, it is necessary to modify the setting of the assimilation experiments.

Thirdly, the analysis SSH RMSEs in all assimilation experiments are larger than the prescribed observation errors. According to the data assimilation theory, analysis RMSEs should be smaller than both background and observation errors. Consequently, the results appear to be inconsistent with established data assimilation theory.

Fourthly, this study aims to compare the impacts of 2 swath and 12 nadir altimeters. However, most spatial maps mainly compare the accuracy of the 2 swath and 12 nadir altimeters with 3 nadir altimeters, which is not aligned with the stated motivation.

Fifthly, this study does not conduct any statistical tests to compare the accuracy of the assimilation experiments.

Finally, this paper lacks the necessary information, particularly regarding model configuration and setup of data assimilation experiments. This deficiency is likely to be attributed to the similarities of the previous authors' work. In addition, numerous colloquial and unclear descriptions as well as typos are found. Therefore, I highly recommend using an English editing service.

Due to these reasons, significant modifications are required before publication. I recommend "major revision" or "reject with an option for resubmission" if the authors require additional time to revise the manuscript.

[Specific comments]

**1) "performance", "quality", and "error": The first two terms are used in a variety of contexts, such as "performance" for computation and "quality" for water quality and statistics. "error" is defined as an instantaneous difference between state variable and true value (e.g., forecast and analysis error: $\epsilon^f = x^f - x^t$ and $\epsilon^a = x^a - x^t$ respectively) and it does not indicate statistic expectations such as RMSE. Therefore, to ensure precision in communication and avoid readers' misunderstanding, it would be more appropriate to use the statistic terminologies "accuracy" and "RMSE" throughout the manuscript instead of these three words.**

**2) Line 29 (L29) in Page 1 (P1): Please clarify the meaning of "blue/white/green ocean states".**

**3) The short introduction is not inadequate, but it lacks the depth needed to fully motivate the comparison of the impacts between the constellation of 2 wide-swath and 12 nadir altimeters. In particular, this paper closely resembles the previous work by the authors (Benkiran et al. 2022), except for adding an experiment assimilating 12 nadir altimeters. While the OSSE allows for the evaluation of observation networks in virtual space by changing the number and type of satellites before their real-world establishment, this paper lacks clear differences in terms of novelty from the previous paper.**

To enhance the novelty of this paper, additional assimilation experiments would be beneficial to understand how many nadir altimeters can surpass the accuracy achieved by 1 or 2 wide-swath altimeters. Furthermore, considering discussions on the costs associated with the development and launch of these satellites would greatly strengthen the paper. By incorporating such discussions, this paper could offer valuable insights and guidance for future satellite missions.

**4) L15 in P2: Please clarify the meaning of "Phase A study".**

**5) The authors frequently rely on references to previous works (e.g., Benkiran et al. 2021, 2022) without providing sufficient details in this paper. It is crucial to include essential information within the context of this study to avoid readers having to refer to multiple papers. Specifically, details on the WiSA concept should be thoroughly described in Section 2, while the model configuration and assimilation settings should be specified in Section 3.**

Please also consider integrating Section 2 into Section 1 to provide a smoother flow of information.

In Section 3, please provide detailed information on the model configuration, including vertical resolution, initial conditions, spin-up period, boundary forcing, and treatment of sea ice if the system is global, differences in configurations and forcings between NEMO versions 3.1 and 3.6, and the settings of the data assimilation experiments.

**6) L21 in P2: Please specify the "WISA #A orbit".**

**7) L31in P2: Please remove "free" because the free run refers to a simulation independent of the Nature Run. It would be better to add descriptions of the free run in the first paragraph of subsection 3.1 rather than subsection 3.4 to clarify the differences between the free and Nature runs.**

**8) The first paragraph in subsection 3.2: Please add the information on observation errors for each variable.**

**9) The second paragraph in subsection 3.2: It is essential to show the SSH observation errors of swath altimeters. If the errors undergo spatiotemporal variations, please present the spatial pattern and average over the global ocean.**

**10) L14 and others: Please remove the space between "Figure 1" and "A" and use a consistent format to indicate figures (e.g. Figure 1a) rather than ambiguous descriptions such as "right figure".**

**11) The last paragraph in subsection 3.2: Please indicate the coverage ratios of each assimilated observation.**

**12) L20 in P3: Please add the information on what data assimilation scheme the SAM2 is based on. If the variational method is used in this study, please specify the details of the prescribed background errors.**

**13) L22 in P3: Please specify "adaptivity scheme" and "observation residuals".**

**14) L31 in P3: Please specify "uncontrolled temporal frequencies".**

**15) L34 in P3: We do not have access to true values in real world and cannot initiate simulations from initial conditions based on true values. In the OSSE, both the free run and data assimilation experiments cannot employ true values from the Nature Run. However, this study uses true values from the Nature Run as the initial conditions for all assimilation experiments. Therefore, the procedure in this study deviates from the typical OSSE setup.**

**16) Subsection 3.4: Please specify the reasons why only the Sentinel-6 is assimilated for all data assimilation experiments.**

**17) L9-L18 in P4: The detailed information on validation methods should not be included in the result section. Please describe it by inserting the method section between Sections 3 and 4.**

**18) L15–17 in P4: Please clarify the reasons why the cutoff scale is defined as 200 and 500 km in this study. The scale of ocean eddies is about 100 km. If the authors intend to separate the meso- and large-scale, the inclusion of the latter one (i.e., 500 km) might not be necessary. Please specify what kind of filters are applied to what values (state vector, error, and RMSE).**

**19) L24-25 in P4: Please specify how the Nature Run and free run reproduce the SSH variance compared with the observations, although the authors might have described the detail in the previous paper. This information is important to demonstrate the reproducibility of the different versions of NEMO.**

**20) L27 in P4 and others: "temporal evolution" indicates time differentiation, d/dt, and is not appropriate to use here and in similar descriptions. "time series" would be a better expression to be consistent with the authors' intent.**

**21) Section 4: To validate the results, this study mainly used two statistics: RMSEs and variance**

errors. For consistency with the dimension of the prescribed background and observation errors (cf. Farchi and Bocquet 2018), it would be more appropriate to use RMSEs rather than variance errors.

**22) L31 in P4 and others: Please summarize the names of data assimilation experiments in subsection 3.4.**

**23) Section 4: As described in subsection 3.2, the prescribed observation errors of nadir altimeters are 2 cm (i.e., 4 cm$^2$ in variance), which are smaller than the best analysis variance errors of 9.6 cm$^2$. Therefore, the results of the assimilation experiments are inconsistent with the data assimilation theory. Please specify the reasons why the variance errors are larger than the prescribed observation errors.**

**24) Section 4: Although the authors showed the time series of variance errors, it is necessary to apply statistical tests, such as paired samples t-test, to detect the significant RMSE differences among all experiments, especially between the 2 swath and 12 Nadir experiments.**

**25) Section 4: Most of the spatial pattern figures showed the statistics differences between 3 Nadirs and 12 Nadirs and between 3 Nadirs and 2 Swath, with the reference defined as 3 Nadirs. However, the main motivation of this paper is to compare the impacts of 2 swath and 12 nadir altimeters. Therefore it is cruicial to show the RMSE differences between 2 swath and 12 nadir altimeters. To detect significant differences between 2 swaths and 12 nadir altimeters, it would be necessary to apply the statistical test to each grid cell.**

**26) L42 in P4: It would be better to calculate areas than points. Please calculate degradation areas as well.**

**27) L24 in P5: Please specify "a variance preserving form".**

**28) Subsection 4.2: It would be expected that 2 swath altimeters would have larger impacts, especially on mesoscale than 12 Nadir altimeters. However, the accuracy differences are substantial at spatial and time scales longer than 200 km and 10 days, respectively (Figs. 8 and 10). It is essential**

to understand the role of altimeters by investigating the causes of why the impacts are substantial at the longer scale only.

**29) L30-32 in P5: Please specify the method used to calculate time spectral coherence (i.e., correlation coefficient). Is it a spatially averaged correlation coefficient? Furthermore, please clearly define "effective temporal resolution".**

**30) The description style is more colloquial throughout the manuscript, especially after the fourth paragraph in subsection 4.2. For example, the sentence "On the right-hand figure, we have the difference between 12 Nadirs and 3 Nadirs" in L35-36 in P5 might be understood by the readers. However, for scientific journals, this colloquial and subjective style is not appropriate, and it is necessary to make objective descriptions. Therefore, I highly recommend using an English editing service.**

**31) The fifth paragraph in subsection 4.2: The Kuroshio and Kuroshio Extension regions would have almost the same order of dynamic energy (i.e., the sum of mean and eddy kinetic energy) as the Gulf Stream region. Since the energy spectra would exclude the mean kinetic energy, the description of the "less energetic Kuroshio region" in L47 in P5 is not reasonable.**

To clarify the different impacts of 2 swath and 12 nadir altimeters, it is essential to investigate what phenomena are reproduced with different accuracies.

**32) L2 in P6 and others: "mean of … error" is a bias? Please use appropriate terminologies.**

**33) L3 in P6: "error" is not "innovation".**

**34) The second paragraph in subsection 4.3, subsection 4.4: The detailed descriptions of validation of the velocities in the second paragraph in subsection 4.3 and particle tracking in subsection 4.4 would not be necessary because the results are largely consistent with the SSH.**

**35) L18 in P6: "significant" can be used only when statistical tests are performed.**

**36) L24-28 in P7: The meaning of the descriptions is unclear. Please carefully revise and clarify the descriptions.**

**37) Section 5: The results from the OSSE in virtual may not necessarily align with the OSE in real. It would be important to inform readers that the validation results might not be consistent between OSSE and OSE due to various factors such as model biases.**

**38) The OSSE indicates a method used to evaluate the impacts of observation networks and does not involve data assimilation experiments. Consequently, "NatRun-OSSEs" in Fig. 3 and similar expressions may not appropriate.**

**39) Figure 8: The correlation coefficient between free run and nature run appears to be completely zero. Even in the chaotic Gulf Stream regions, the coefficient would not be zero. Please carefully review the calculation.**

**40) There is no label on y-axis in Fig. 8a, y-axis of Fig. 10a, and color scales of Fig. 11.**

[Technical corrections]

Line 1 (L1) in Page 1 (P1): "for" in the Title might be "on".

L9 in P1: "Surface Water Ocean Topography" might be "Surface Water Ocean Topography (SWOT) mission".

L15 in P1: "has" should be "have". Hereafter, we use right arrows → for similar replacement.

L18 in P1: Insert "The results showed that" before "These two configurations" to clarify that the results performed by the authors are shown hereafter.

L35 in P1: "plays" → "play".

L15 in P2: Please spell out "CNES".

L31-32 in P2: "to represent … synthetic observations" might be "to generate true values and synthetic observations". It would be better to use "generate observations" rather than "simulate observations" in the OSSE.

L39 in P2: Remove "same".

L40-43 in P2: Please specify satellite and in-situ observations for each variable.

L44 in P2: "simulated" should be "distributed". Remove "along" before "the swath".

L45 in P2: Please clarify "(S1 and S2)".

L1 in P3: "(S1, S2)" → "(S1 and S2)".

L6 in P3: "but" should be inserted before "separated".

L28 in P3: "injected" → "inserted".

L39 in P3-L2 in P4 and others: The caption of tables should be located at the top of the tables.

L40 in P3: "Column" → "column"

L41 in P3: "Column 2" → "the second column", and "standard" → "assimilated".

L15 in P4: "of high variability"→ "with large variations".

L18 in P4: "depth" → "ocean interior"

L18 in P4: Please clarify "system mass".

L31 in P4: "smaller" should be inserted before "SSH error".

L6-7 in P5: Please clarify what experiments are compared.

L21-22 in P5: Please check the descriptions.

L28 in P5: Which scale "onwards" indicates, longer or shorter?

L32 in P5: "wavelengths" → "timescale"

L27-28 in P6: Please specify which way the particle tracking is conducted, offline or online.

L3 in P10: "Truth Run" → " Nature Run"

Figure 2: The color scale should be modified because there are no minus values. Units with a color scale "$Cm^2$" should be "$cm^2$", and this is the same for others.

Figure 3: The color of forecast error variance is difficult to see. Please modify the line type and color.

L9 in P12: The descriptions of the fourth and fifth columns are not consistent with the Table 2.

Label of Fig. 7: Please clarify which RMSE is shown in Fig. 7, forecast or analysis RMSEs. Please clarify which data assimilation experiments are subtracted from which experiments.

Figure 8b: The unit in the label of x-axis "day" should be "$day^{-1}$".

L8 in P14: "cm2" → "$cm^2$"

Labels of Figs. 9 and 10: These are almost same as Figs. 7 and 8. Please use the expression "As in … but for …".

===Reference

Farchi and Bocquet (2018), **25**, 4, 765-807, Nonlinear Processes in Geophysics

---

## Author Comment (AC1)

[General comments]

This paper investigates the impacts of two swath and 12 nadir altimeters on the analysis and forecast accuracies conducting the OSSEs. The topic appears to be interesting, however, there are critical issues described below.

Firstly, the contents are largely similar to those of a previous paper by Benkiran et al. (2022), but with only the addition of the results assimilating 12 nadir altimeters. To establish the novelty of this paper, I strongly suggest conducting a wider variety of experiments to comprehensively investigate how many nadir altimeters are beyond 1- and 2-swath altimeters and cannot substantially increase the accuracy.

Secondly, in the OSSE, data assimilation experiments should not employ true values from the Nature Run. However, this study uses true values for the initial conditions of assimilation experiments, which is inconsistent with the OSSE protocols. Therefore, it is necessary to modify the setting of the assimilation experiments.

Thirdly, the analysis SSH RMSEs in all assimilation experiments are larger than the prescribed observation errors. According to the data assimilation theory, analysis RMSEs should be smaller than both background and observation errors. Consequently, the results appear to be inconsistent with established data assimilation theory.

Fourthly, this study aims to compare the impacts of 2 swath and 12 nadir altimeters. However, most spatial maps mainly compare the accuracy of the 2 swath and 12 nadir altimeters with 3 nadir altimeters, which is not aligned with the stated motivation.

Fifthly, this study does not conduct any statistical tests to compare the accuracy of the assimilation experiments.

Finally, this paper lacks the necessary information, particularly regarding model configuration and setup of data assimilation experiments. This deficiency is likely to be attributed to the similarities of the previous authors' work. In addition, numerous colloquial and unclear descriptions as well as typos are found. Therefore, I highly recommend using an English editing service.

Due to these reasons, significant modifications are required before publication. I recommend "major revision" or "reject with an option for resubmission" if the authors require additional time to revise the manuscript.

The content of this article is quite different from the Benkiran et al (2022) paper for the following reasons:
- The 12 nadirs of these experiments are used in SAR mode in the same (orbital) plane as Sentinel-3. The Nadir error is smaller (2 cm).
- The Karin errors used for the swath altimeters (WiSa) fully take into account the effect of waves on the measurement noise as cited in the paper, whereas in the paper by Benkiran et al (2022) a 2 m constant wave height was considered in the observation simulator.

The reviewer misunderstood our OSSE protocol which is a rigorous and state-of-the-art one. In our OSSEs, the Natrun is only used to simulate observations (to which a measurement noise is added) and for validation. As described in the paper (Benkiran et al. 2021), the initial state of our simulations is very different from NatRun. This OSSE protocol has been fully validated in this paper. The following figure shows the variance difference between NatRun and the free simulation used for our initial state. We will better explain the OSSE protocol in the paper to avoid any misunderstanding.

[Figure]

*Figure 1:* SSH Variance error for Free Run (Cm²)

[Specific comments]

**1) "performance", "quality", and "error": The first two terms are used in a variety of contexts, such as "performance" for computation and "quality" for water quality and statistics. "error" is defined as an instantaneous difference between state variable and true value (e.g., forecast and analysis error: $\epsilon f = xf - xt$ and $\epsilon a = xa - xt$ respectively) and it does not indicate statistic expectations such as RMSE. Therefore, to ensure precision in communication and avoid readers' misunderstanding, it would be more appropriate to use the statistic terminologies "accuracy" and "RMSE" throughout the manuscript instead of these three words.**

We have revised the use of performance, quality and error terms in the paper and clarified what we meant when needed

**2) Line 29 (L29) in Page 1 (P1): Please clarify the meaning of "blue/white/green ocean states".**

Blue = physics, White = sea ice, Green = biogeochemistry

**3) The short introduction is not inadequate, but it lacks the depth needed to fully motivate the comparison of the impacts between the constellation of 2 wide-swath and 12 nadir altimeters. In particular, this paper closely resembles the previous work by the authors (Benkiran et al. 2022), except for adding an experiment assimilating 12 nadir altimeters. While the OSSE allows for the evaluation of observation networks in virtual space by changing the number and type of satellites before their real-world establishment, this paper lacks clear differences in terms of novelty from the previous paper.**

We better explained the novelty of the paper in the introduction. More realistic simulations that better take into account the Karin noise and comparison of the 2 scenarios discussed for evolution of the Copernicus Sentinel 3 mission (Sentinel 3 New Generation Topo)

To enhance the novelty of this paper, additional assimilation experiments would be beneficial to understand how many nadir altimeters can surpass the accuracy achieved by 1 or 2 wide-swath altimeters. Furthermore, considering discussions on the costs associated with the development and launch of these satellites would greatly strengthen the paper. By incorporating such discussions, this paper could offer valuable insights and guidance for future satellite missions.

We understand that it would be interesting to include a discussion on cost issues but the information is not available (there are on going industrial studies managed by ESA on this topic but information is not shared publicly). Our paper focused on the relative technical performance of the two concepts.

**4) L15 in P2: Please clarify the meaning of "Phase A study".**

Phase A study means feasibility and preliminary design study. This is not precise in the paper.

**5) The authors frequently rely on references to previous works (e.g., Benkiran et al. 2021, 2022) without providing sufficient details in this paper. It is crucial to include essential information within the context of this study to avoid readers having to refer to multiple papers. Specifically, details on the WiSA concept should be thoroughly described in Section 2, while the model configuration and assimilation settings should be specified in Section 3.**

Please also consider integrating Section 2 into Section 1 to provide a smoother flow of information. In Section 3, please provide detailed information on the model configuration, including vertical resolution, initial conditions, spin-up period, boundary forcing, and treatment of sea ice if the system is global, differences in configurations and forcings between NEMO versions 3.1 and 3.6, and the settings of the data assimilation experiments.

We have added additional information to avoid readers having to refer to multiple papers.

**6) L21 in P2: Please specify the "WISA #A orbit". The so-called WiSA Phase A orbit was selected by CNES using the methodology of Dibarboure et al. (2018) to maximize the sampling for one- to three-swath altimeter satellites (or swath–nadir hybrid constellations).**

**7) L31 in P2: Please remove "free" because the free run refers to a simulation independent of the Nature Run. It would be better to add descriptions of the free run in the first paragraph of subsection 3.1 rather than subsection 3.4 to clarify the differences between the free and Nature runs.**

The free simulation, as indicated at the beginning of the review, is different from NatRun. This is the model into which we assimilate the data simulated from NatRun.

**8) The first paragraph in subsection 3.2: Please add the information on observation errors for each variable. Adding in article**

**9) The second paragraph in subsection 3.2: It is essential to show the SSH observation errors of swath altimeters. If the errors undergo spatiotemporal variations, please present the spatial pattern and average over the global ocean. I adde Swath's observation error profile as a function of the Waves in the article (figure 1).**

**10) L14 and others: Please remove the space between "Figure 1" and "A" and use a consistent format to indicate figures (e.g. Figure 1a) rather than ambiguous descriptions such as "right figure". Adding in article**

**11) The last paragraph in subsection 3.2: Please indicate the coverage ratios of each assimilated observation. ". Adding in article**

**12) L20 in P3: Please add the information on what data assimilation scheme the SAM2 is based on. If the variational method is used in this study, please specify the details of the prescribed background errors.**

Yes, we use the SAM2 assimilation system (based on SEEK, developed at MOI). A detailed description can be found in Benkiran et al. 2021. The error covariance matrix is constructed from a base of anomalies calculated from a free simulation of a long period.

**13) L22 in P3: Please specify "adaptivity scheme" and "observation residuals". Adaptativity technic…..**

The adaptativity technic is based on the work of Desroziers and Ivanov (2001) and aims to find a scalar for each local region, that multiplies the local restriction of P, so that the following equation is satisfied for each local region and each analysis cycle. As presented in Benkiran et al 2021.

**14) L31 in P3: Please specify "uncontrolled temporal frequencies".**
With our assimilation scheme, we don't control for all the exiting frequencies in the model, which is hard on our analysis frequency and forecast error covariance matrices.

**15) L34 in P3: We do not have access to true values in real world and cannot initiate simulations from initial conditions based on true values. In the OSSE, both the free run and data assimilation experiments cannot employ true values from the Nature Run. However, this study uses true values from the Nature Run as the initial conditions for all assimilation experiments. Therefore, the procedure in this study deviates from the typical OSSE setup.**

In our study, NatRun is very different from the model in which we assimilate simulated data from NatRun, as shown above. The first figures (Fig1 - Fig5) in the paper by Benkiran et al 2021 show this difference. We have a significant difference in Ssh, KE (Kinetic Energy, 2015 mean), mixing layer depth and surface temperature.

**16) Subsection 3.4: Please specify the reasons why only the Sentinel-6 is assimilated for all data assimilation experiments.**

Other altimetry satellites are likely to produce data at the same time as S3-NG and Sentinel-6. We focus only on Sentinel altimeters, with a better knowledge and accuracy of Sentinel-6 (Jason-3: which is representative of Sentinel-6).

**17) L9-L18 in P4: The detailed information on validation methods should not be included in the result section. Please describe it by inserting the method section between Sections 3 and 4. update in the article**

**18) L15–17 in P4: Please clarify the reasons why the cutoff scale is defined as 200 and 500 km in this study. The scale of ocean eddies is about 100 km. If the authors intend to separate the meso- and large-scale, the inclusion of the latter one (i.e., 500 km) might not be necessary. Please specify what kind of filters are applied to what values (state vector, error, and RMSE).**

**19) L24-25 in P4: Please specify how the Nature Run and free run reproduce the SSH variance compared with the observations, although the authors might have described the detail in the previous paper. This information is important to demonstrate the reproducibility of the different versions of NEMO.**

**20) L27 in P4 and others: "temporal evolution" indicates time differentiation, d/dt, and is not appropriate to use here and in similar descriptions. "time series" would be a better expression to be consistent with the authors' intent. Update in article**

**21) Section 4: To validate the results, this study mainly used two statistics: RMSEs and variance errors. For consistency with the dimension of the prescribed background and observation errors (cf. Farchi and Bocquet 2018), it would be more appropriate to use RMSEs rather than variance errors. ok**

**22) L31 in P4 and others: Please summarize the names of data assimilation experiments in subsection 3.4. Update in article**

**23) Section 4: As described in subsection 3.2, the prescribed observation errors of nadir altimeters are 2 cm (i.e., 4 cm2 in variance), which are smaller than the best analysis variance errors of 9.6 cm2. Therefore, the results of the assimilation experiments are inconsistent with the data assimilation theory. Please specify the reasons why the variance errors are larger than the prescribed observation errors.**

As show in figure 5(A) (SSH variance error for 3Nadirs experiment), we have a regional error variance that depends on local variability. We have larger errors than the instrument error in the western currents, the circumpolar current and the barotropic regions. The figures compared in the paper represent global averages. In addition to the observation error, we used an SSH representativity error (Figure 8, Benkiran et al. 2021) this representativity error for the SSH observations that were calculated from the standard deviation of the NatRun SSH for scales smaller than 100 km

**24) Section 4: Although the authors showed the time series of variance errors, it is necessary to apply statistical tests, such as paired samples t-test, to detect the significant RMSE differences among all experiments, especially between the 2 swath and 12 Nadir experiments.**

**25) Section 4: Most of the spatial pattern figures showed the statistics differences between 3 Nadirs and 12 Nadirs and between 3 Nadirs and 2 Swath, with the reference defined as 3 Nadirs. However, the main motivation of this paper is to compare the impacts of 2 swath and 12 nadir altimeters. Therefore, it is cruicial to show the RMSE differences between 2 swath and 12 nadir altimeters. To detect significant differences between 2 swaths and 12 nadir altimeters, it would be necessary to apply the statistical test to each grid cell.**

**26) L42 in P4: It would be better to calculate areas than points. Please calculate degradation areas as well. update in the article**

**27) L24 in P5: Please specify "a variance preserving form".**

Variance-preserving spectra are designed to provide a useful measure of the signal variance. **Emery and Thomson** provide a succinct derivation of this. If you plot the spectrum Sxx times frequency f , as a function of log(f ), then area under the spectral curve between frequencies f0 and fn will be:

$$\sigma^2 = \int_{f_0}^{f_n} f S_{xx}(f) d(\log(f)) = {}_0 + \int_{f_0}^{f_n} S_{xx}(f) df$$

Here note that d(log(f )) = df /f . Thus area under the curve between two frequencies gives a measure of spectral signal variance in that frequency band

**28) Subsection 4.2: It would be expected that 2 swath altimeters would have larger impacts, especially on mesoscale than 12 Nadir altimeters. However, the accuracy differences are substantial at spatial and time scales longer than 200 km and 10 days, respectively (Figs. 8 and 10). It is essential to understand the role of altimeters by investigating the causes of why the impacts are substantial at the longer scale only.**

Yes, we tested different separation cuts to highlight the impact, we have this set (200km and 500km) because that's where the impact is clearest. We used a Lanczos low-pass filter (python) to separate these scales.

**29) L30-32 in P5: Please specify the method used to calculate time spectral coherence (i.e., correlation coefficient). Is it a spatially averaged correlation coefficient? Furthermore, please clearly define "effective temporal resolution".**

Yes, Is it a spatially averaged correlation coefficient. Update effective temporal resolution in article.

**30) The description style is more colloquial throughout the manuscript, especially after the fourth paragraph in subsection 4.2. For example, the sentence "On the right-hand figure, we have the difference between 12 Nadirs and 3 Nadirs" in L35-36 in P5 might be understood by the readers. However, for scientific journals, this colloquial and subjective style is not appropriate, and it is necessary to make objective descriptions. Therefore, I highly recommend using an English editing service.**

we used an English-language publishing service.

**31) The fifth paragraph in subsection 4.2: The Kuroshio and Kuroshio Extension regions would have almost the same order of dynamic energy (i.e., the sum of mean and eddy kinetic energy) as the Gulf Stream region. Since the energy spectra would exclude the mean kinetic energy, the description of the "less energetic Kuroshio region" in L47 in P5 is not reasonable.**

To clarify the different impacts of 2 swath and 12 nadir altimeters, it is essential to investigate what phenomena are reproduced with different accuracies.

**32) L2 in P6 and others: "mean of … error" is a bias? Please use appropriate terminologies. Update in article**

**33) L3 in P6: "error" is not "innovation". Error**

**34) The second paragraph in subsection 4.3, subsection 4.4: The detailed descriptions of validation of the velocities in the second paragraph in subsection 4.3 and particle tracking in subsection 4.4 would not be necessary because the results are largely consistent with the SSH.**

**35) L18 in P6: "significant" can be used only when statistical tests are performed. Update in article**

**36) L24-28 in P7: The meaning of the descriptions is unclear. Please carefully revise and clarify the descriptions.**

**37) Section 5: The results from the OSSE in virtual may not necessarily align with the OSE in real. It would be important to inform readers that the validation results might not be consistent between OSSE and OSE due to various factors such as model biases.**

**38) The OSSE indicates a method used to evaluate the impacts of observation networks and does not involve data assimilation experiments. Consequently, "NatRun-OSSEs" in Fig. 3 and similar expressions may not appropriate. update in the article**

**39) Figure 8: The correlation coefficient between free run and nature run appears to be completely zero. Even in the chaotic Gulf Stream regions, the coefficient would not be zero. Please carefully review the calculation.**

**40) There is no label on y-axis in Fig. 8a, y-axis of Fig. 10a, and color scales of Fig. 11. update in the article**

[Technical corrections]

Line 1 (L1) in Page 1 (P1): "for" in the Title might be "on". update in the article

L9 in P1: "Surface Water Ocean Topography" might be "Surface Water Ocean Topography (SWOT) mission". update in the article

L15 in P1: "has" should be "have". Hereafter, we use right arrows -> for similar replacement. update in the article

L18 in P1: Insert "The results showed that" before "These two configurations" to clarify that the results performed by the authors are shown hereafter.

L35 in P1: "plays" à "play". update in the article

L15 in P2: Please spell out "CNES". update in the article

L31-32 in P2: "to represent … synthetic observations" might be "to generate true values and synthetic observations". It would be better to use "generate observations" rather than "simulate observations" in the OSSE. update in the article

L39 in P2: Remove "same". Distributed update in the article

L40-43 in P2: Please specify satellite and in-situ observations for each variable.

> For altimeters, the names of the satellites used are given in the following paragraph (3Nadirs: Sentinel 6A, Sentinel 3A and Sentinel 3B, Table 1),
> The temperature and salinity profiles were extracted at the same points and dates as the real in-situ profiles observed in 2015 and found in the CORA4.1 database stored in the Coriolis and CMEMS in-situ data centre (Cabanes et al. 2013)
> Cabanes, C., Grouazel, A., von Schuckmann, K., Hamon, M., Turpin, V., Coatanoan, C., et al. (2013). The CORA dataset: validation and diagnostics of in-situ ocean temperature and salinity measurements. Ocean Science 9, 1–18. doi:https://doi.org/10.5194/os-9-1-2013.

L44 in P2: "simulated" should be "distributed". Remove "along" before "the swath". update in the article

L45 in P2: Please clarify "(S1 and S2)". update in the article

L1 in P3: "(S1, S2)" → "(S1 and S2)". update in the article

L6 in P3: "but" should be inserted before "separated". update in the article

L28 in P3: "injected" → "inserted". update in the article

L39 in P3-L2 in P4 and others: The caption of tables should be located at the top of the tables. update in the article

L40 in P3: "Column" →"column". update in the article

L41 in P3: "Column 2" → "the second column", and "standard" → "assimilated". update in the article

L15 in P4: "of high variability"→ "with large variations". update in the article

L18 in P4: "depth"→ "ocean interior" update in the article

L18 in P4: Please clarify "system mass".

L31 in P4: "smaller" should be inserted before "SSH error". update in the article

L6-7 in P5: Please clarify what experiments are compared.

L21-22 in P5: Please check the descriptions.

L28 in P5: Which scale "onwards" indicates, longer or shorter?

L32 in P5: "wavelengths" → "timescale"   update in the article

L27-28 in P6: Please specify which way the particle tracking is conducted, offline or online.

L3 in P10: "Truth Run" → " Nature Run" update in the article

Figure 2: The color scale should be modified because there are no minus values. Units with a color

scale "Cm2" should be "cm2", and this is the same for others. Yes, I've corrected the legend (cm$^2$), it's a variance so it's not negative.

Figure 3: The color of forecast error variance is difficult to see. Please modify the line type and color. update in the article

L9 in P12: The descriptions of the fourth and fifth columns are not consistent with the Table 2. Corrected in the manuscript

Label of Fig. 7: Please clarify which RMSE is shown in Fig. 7, forecast or analysis RMSEs. Please clarify which data assimilation experiments are subtracted from which experiments.

Our study consists of OSSE, we have the complete fields on the grid from which we simulated our observations (NatRun, our reality), here we show: figure 1, the SSH Rmse between the NatRun and the experiment with the assimilation of 3Nadirs. The figure represents the SSH Rmse between the experiment with 3Nadirs and 12 Nadirs and the last one between the experiment with 2Swaths and 3Nadirs.

Figure 8b: The unit in the label of x-axis "day" should be "day-1". update in article

L8 in P14: "cm2" à "cm2" update in article

Labels of Figs. 9 and 10: These are almost same as Figs. 7 and 8. Please use the expression "As in … but for …".update in article

---

## Author Comment (AC2)

This article assesses the impact of the sea surface height (SSH) observation for the global ocean data assimilation system. The observation system simulation estimations (OSSEs) are performed and compared the impact of constellation of two wide-swath altimeters and twelve nadir altimeters to the current operating three nadir altimeters. Nature run for the OSSEs is obtained from the results without assimilation based on the latest operational model configurations, while the OSSEs uses the previous version of the model settings. The OSSEs show that the error reduction of SSH in the 2Swaths experiment is larger than other experiments. The impacts are also evaluated through the error reduction of the temperature and salinity profiles and surface velocity. The satellite constellation proposed in this study will be a next generation for the altimetry mission. Thus, the OSSEs have significant role on the decision of the future mission. The article almost satisfies the quality for publication, but there are several issues and concerns which should be improved. The detail is described below.

Major comments

Please specify the background SSH variance from Nature Run in the Fig. 6. It can be a lead time of the SSH forecast. For example, if the background SSH variance is about 20 cm$^2$, the lead time is about 6 days (5days) in the 2Swaths (12Nadirs) experiment. Firstly, I expected that the large error reduction in the western boundary regions led to the more improvement of the separation distance (Figure. 16 B and C). However, the results show that the improvement is greater in the low latitudes not in the western boundary regions. Is it related to lead time for the OSSEs?

Figure shows the SSH variance of NatRun, we also show (as in Figure 6) the 7-day forecast score in 3 different regions (with signals of different frequencies). We note that the impact of the 2 Swaths is also significant over regions of high variability such as the Gulf Stream and less significant over the Kuroshio area as mentioned in our paper.

[Figure]

*Figure 1: SSH variance (in cm2) in the NatRun over the period from February to December 2015. Variance of the error for each day of forecast (7 days, cm$^2$) considering the SSH in Kuroshio, Guls Stream and North Atlantic Drift areas*

.

In Figure. 15, the surface velocities in the low latitudes have been improved. However, the additional SSH impact in the low latitudes is less effective compared to the middle and high latitude regions as shown in Figure 4 and 5. Why the surface velocities in the low latitudes are improved? There are two possible reasons. One is that the impact of SSH assimilation just on the low latitudes directly improves surface velocities. The other is that the SSH improvement in subtropical regions have adjusted the dynamical balance in the equator. Which is the more plausible reason?

Yes, the reduction in the error with the 2Swaths is more marked on surface velocities in low latitudes and subtropical regions for two reasons:

- Surface velocities control by SSH assimilation (your first assumption)
- with the assimilation of the 12 nadirs, the remaining error is already too small.

With respect to the surface velocity, there is another concern. Tchonang et a., 2021 shows that the OSSE with single swath data had negative impact on the zonal velocity error in the equator in their Figure 12. However, Figure 15 in this study shows the improvement of the surface velocities in the low latitude. Does it mean that the more observation data obtained from the altimeter constellation can explain the improvement?

Anyway, there seems to be no explanation and discussion about Figure 15 in the manuscript. Please also check this point.

Yes, with SWOT data assimilation (1Swath with a 21-day cycle) the impact on zonal velocities at the equator was negative. This is essentially due to the coverage of SWOT data over one analysis cycle (7 days) given that we are in a region with fast signals. With 2Swaths, however, we have good coverage and better control of these fast signals.

How many SSH data are used for assimilation? Could you describe the number of SSH observation in the manuscript with respect to Figure 1? Which has more observations in Fig.1 C or D? As described in the manuscript, there is more improvement in 2Swaths run than 12Nadirs run. It will be related with the difference in the number of SSH data and/or the spatial area available for observation (swath v.s. nadir). Which is more efficient for reducing the SSH error?

In fact, the number of observations per analysis cycle (7 days) is very high. The figure below shows the number of observations per analysis cycle, in black for the 3Nadirs, blue for the 12Nadirs and red for the 2 Wide-Swaths. There is a factor of 3.5 between the number of 2Wide-Swaths observations and the 12Nadirs.

[Figure]

[Figure]

[Figure]

Figure 2 Observations Number by assimilation cycle (7days)

Figure 3: Observations Number by latitude during 7days

Figure 7 shows the interesting result, indicating the SSH variations in the Gulf Stream area is difficult to control by SSH assimilation comparing to the Kuroshio extension region shown in Figure 9. The SSH RMS error of the constellation experiments is almost similar to the 3Nadir experiment. What caused the comparable SSH error in early April for 2Swaths and in early July for 12Nadirs? Is it related with the position of the Gulf Stream axis?

Yes, indeed, SSH control is very difficult by assimilation over the gulf stream region because of the position of the stream and the frequency of the signals present in this area, as shown in figure 12 (wavenumber-frequency) compared to the Kuroshio area.

For figure 7, I looked in more detail at this point on the peaks on the SSH rmse (April, July and October...). I found the problem which comes from the residues of the fast signals in our NatRun (Benkiran at al. 2021). on the following figure (example over the period from 08/07 to 28/07/2015) we show the impact of these residues on the yellow sea (120°E-130°E). we have redone the same statistics by deleting this zone.

[Figure]

Minor comments

P2 L44: Benkiran et al., (2012) is (2021) or (2022)? update in the article

P3 L3: A random noise is set to 2 cm in the manuscript, but 3cm in the previous studies (e.g., Benkiran et al., 2021, 2011). Is this correct?

Yes, in this study we use Nadirs data in SAR mode, which is why we have them with a smaller error of 2cm.

P3 L19: The article focuses on the impact of the SSH assimilation. Therefore, it is better to describe the method of the assimilation scheme, especially for SSH variable. In Benkiran et al., (2021), SEEK filter, which is one of the sequential assimilation schemes, for the short-term variations is used for assimilation scheme. It will be easier for the readers to find out the forecast and analysis cycles. In addition, how was the mean surface height for the ocean model obtained? The mean surface height has a critical role of the SSH assimilation.

P5 L34: If the authors deal with the black box area in Figure. 7A, it is better to use "Kuroshio extension" to specify the area. update in the article

P6 L36 "Figures 15 B and C" is "Figures 16 B and C" update in the article

P8-9: Please check the order of the references. Sometimes the order should be reverse such as Vergara et al., 2019 and Ubelmann et al., 2015. update in the article

P12 Table2: The SSH variance error for wavelengths smaller than 500km in the 2Swatths experiment is 84 in Table 2. The value seems to be wrong. Sorry, typing error (8.4)

P14 & P15 Fig. 7 D and Fig. 9 D: Why does the figure start from February? update in the article

P15 Fig. 10: The title of the left figure is (A)? update in the article

P18 Fig. 16 and 17: Please add (A), (B), (C), etc in the figure title. update in the article